# Improvement of the Structure and Physicochemical Properties of Polylactic Acid Films by Addition of Glycero-(9,10-trioxolane)-Trialeate

**DOI:** 10.3390/polym14173478

**Published:** 2022-08-25

**Authors:** Olga Alexeeva, Anatoliy Olkhov, Marina Konstantinova, Vyacheslav Podmasterev, Ilya Tretyakov, Tuyara Petrova, Olga Koryagina, Sergey Lomakin, Valentina Siracusa, Alexey L. Iordanskii

**Affiliations:** 1N.M. Emanuel Institute of Biochemical Physics, Russian Academy of Sciences, 119334 Moscow, Russia; 2N.N. Semenov Federal Research Center for Chemical Physics, Russian Academy of Sciences, 119991 Moscow, Russia; 3Academic Department of Innovational Materials and Technologies Chemistry, Plekhanov Russian University of Economics, 117997 Moscow, Russia; 4Department of Chemical Science (DSC), University of Catania, Viale A. Doria 6, 95125 Catania, Italy

**Keywords:** polylactic acid, polymer films, ozonide, trioleate, biocompatibility

## Abstract

Glycero-(9,10-trioxolane)-trioleate (ozonide of oleic acid triglyceride, OTOA) was introduced into polylactic acid (PLA) films in amounts of 5, 10, 30, 50, and 70% *w*/*w*. The morphological, mechanical, thermal, and water absorption properties of PLA films after the OTOA addition were studied. The morphological analysis of the films showed that the addition of OTOA increased the diameter of PLA spherulites and, as a consequence, increased the proportion of amorphous regions in PLA films. A study of the thermodynamic properties of PLA films by differential scanning calorimetry (DSC) demonstrated a decrease in the glass transition temperature of the films with an increase in the OTOA content. According to DSC and XRD data, the degree of crystallinity of the PLA films showed a tendency to decrease with an increase in the OTOA content in the films, which could be accounted for the plasticizing effect of OTOA. The PLA film with 10% OTOA content was characterized by good smoothness, hydrophobicity, and optimal mechanical properties. Thus, while maintaining high tensile strength of 21 MPa, PLA film with 10% OTOA showed increased elasticity with 26% relative elongation at break, as compared to the 2.7% relative elongation for pristine PLA material. In addition, DMA method showed that PLA film with 10% OTOA exhibits increased strength characteristics in the dynamic load mode. The resulting film materials based on optimized PLA/OTOA compositions could be used in various packaging and biomedical applications.

## 1. Introduction

In recent decades, large volumes of non-degradable polymer packaging materials have been produced by petrochemical industry, making significant contribution to the global problem of environmental pollution [1]. One of the promising ways to address this problem is the production of packaging materials using biodegradable polymers [2]. Today, several biodegradable polymers, such as polylactic acid (PLA), polyhydroxybutyrate, and poly(ε-caprolactone) are well known [3] and are widely used in medicine [4], tissue engineering [5], production of disposable tableware and packaging materials [6].

Polylactic acid (PLA) is a compostable polymer derived from natural raw materials, which is synthesized from lactic acid monomers by catalytic ring-opening polymerization [7]. PLA could be formed via electrospinning into ultrafine fiber materials [8], extruded into films and molded into diverse shapes [9]. PLA is one of the most frequently used biobased materials in the food packaging industry, which is applied for production of disposable tableware, vegetables packaging, and fast-food containers [10]. Though PLA is a biodegradable polymer, its degradation can occur only in hydrolytic and enzymatic media [11]. PLA degradation takes place in several stages: diffusion of water into the matrix, hydrolysis of ester bonds, reduction in molecular weight, and absorption of residues by microorganisms. The rate of hydrolysis depends on the temperature and water content and is catalyzed by the free carboxyl groups of the hydrolyzed ends of the PLA [12].

The successful application of biodegradable polymer materials in packaging should be based on good mechanical and barrier properties, improved elasticity, biodegradability, etc. [13]. Widespread use of PLA films in packaging applications is restricted by their poor ductility and barrier properties, relatively low thermostability [14]. Considerable efforts have been made to improve the physicochemical properties of PLA films in order to accelerate their employment in packaging industry, such as the addition of modifiers, copolymerization or blending [15,16]. PLA blending with other polymers, such as poly(hydroxybutyrate) (PHB), could significantly improve the final properties of polymer films [17,18,19]. It should be noted that blending of PLA with other polymers requires the use of plasticizers such as limonene [20] and polyethylene glycol [21], in order to increase the miscibility of the polymers in the blend and provide the necessary ductility. Due to strict requirements in food packaging applications, biocompatible and non-toxic plasticizers should be provided in order to improve the properties of the PLA-based films.

It should be noted that modification of PLA materials with functional additives could not only improve the physicochemical properties of PLA films and fiber materials but could also provide additional functionality for various applications [22]. Inclusion of various antimicrobial substances in the polymer matrix could impart antibacterial properties to the material, prevent the growth of microbes inside the package, and also improve the mechanical properties [23]. The use of packaging materials based on biodegradable PLA with antimicrobial additives can reduce environmental impact and provide product protection from physical, chemical, and microbiological factors, which is of great interest in terms of sustainability, adding functional properties to the packaging materials and reducing environmental risks [24,25].

According to the literature, ozonides of vegetable oils (olive, sunflower, etc.), which are the products of ozone reaction with the C=C double bonds in the molecules of unsaturated fatty acids [26], possess good antibacterial properties and could be used as functional additives for various PLA materials [27]. Moreover, ozonated vegetable oils are even more promising in this respect, since they could combine apparent antimicrobial activity with good plasticizing properties [28,29]. The most promising product of ozonation of the natural vegetable oils is 2lycerol-(9,10-trioxolane)-trioleate (ozonide of oleic acid triglyceride, OTOA) [30]. It is non-toxic, biocompatible, and has good biodegradability and antimicrobial activity. In a recent work [30] it was shown that introduction of OTOA into the nonwoven fibrous PLA materials has a significant effect on their physicochemical and functional properties and could provide additional antimicrobial functionality [31]. Thus, OTOA could be a promising additive for introduction into the PLA film matrix in order to provide additional functionality for various packaging applications.

In this work, we obtained novel film materials based on PLA with the addition of various amounts of OTOA (0–70 wt.%) as a modifier. DSC, DMA, X-ray diffraction analysis, FTIR spectroscopy, and optical and polarization microscopy were used to study the morphological, physicochemical, mechanical, and water absorption properties of PLA films with various OTOA contents. The obtained results provided a deeper understanding of the specific interactions of PLA with OTOA and made it possible to establish the effect of the OTOA content on the physicochemical and functional properties of the obtained film materials.

## 2. Materials and Methods

### 2.1. Materials

PLA was purchased from NatureWorks**^®^** Ingeo™ 3801X Injection Grade from Shenzhen Bright China Inc. (Shenzhen, China) with a viscosity average molecular weight of 1.9 × 10^5^ g/mol; dry-cleaned chloroform, ≥99.5%, Sigma-Aldrich Inc. (St. Louis, MO, USA) was used to prepare solutions; Glycero-(9,10-trioxolane)-trioleate (ozonide of oleic acid triglyceride (OTOA)) (Figure 1) was obtained from Medozon (Moscow, Russia). The chemical structure of OTOA has been described previously [30,32]. All reagents were used as received.

### 2.2. Preparation of Films

All films used in this work were prepared using solvent evaporation. PLA (2 g) was dissolved in chloroform (50 mL). Then, a certain amount of OTOA (5, 10, 30, 50, and 70 wt.%) was added to the PLA solution in chloroform and stirred for 12 h with a magnetic stirrer. After mixing, the solution was poured onto a glass plate and the film formed was dried to constant weight at ambient temperature. A film of the pristine PLA without additive was used as a control sample.

### 2.3. IR Spectroscopy

PLA films were analyzed using IR Fourier analysis with a NETZSCH TG 209 F1 (NETZSCH-Gerätebau GmbH, Selb, Germany) thermoanalytical balance and a Bruker Tensor 27 IR (Billerica, MA, USA) Fourier spectrometer with PIKE MIRacle™ accessory (PIKE Technologies, Madison, WI, USA) equipped with a germanium (Ge) crystal and an ATR attachment with a Teflon cell and cesium antimony electrode, which allows the measurements of solid samples. The sample was placed on the surface of the crystal and tightly clamped to ensure optical contact. IR spectra were recorded in the range of 4000–400 cm^−1^ with a resolution of 4 cm^−1^ and averaging over 16 successive scans.

### 2.4. DSC

The thermophysical properties of the film materials were determined on a NETZSCH DSC 204F1 Phoenix differential scanning calorimeter (NETZSCH-Gerätebau GmbH, Selb, Germany) in an inert atmosphere at an argon flow rate of 100 mL/min. PLA film samples (about 5 mg) were placed in aluminum sample crucibles and heated from 20 °C to 200 °C at a rate of 10 °C/min. The instrument was calibrated against indium, tin, and lead. After the first heating cycle, the samples were kept at 200 °C for 5 min, cooled to room temperature, and then reheated to 200 °C at a rate of 10 °C/min to record the second DSC curve. All samples were tested in triplicate.

The deconvolution of DSC peaks was carried out by means of NETZSCH (NETZSCH-Gerätebau GmbH, Selb, Germany) Peak Separation 2006.01 software employing the nonlinear regression method for the asymmetric DSC curves (Fraser-Suzuki algorithm) [33]. In calculation, the least squares (SLS) reduction was achieved using a hybrid procedure in which the Levenberg/Marquardt method was combined with step length optimization [34].

### 2.5. X-ray Diffraction Analysis

The structure of the PLA and PLA + OTOA films was studied by X-ray diffraction (XRD) using a DRON-3M X-ray diffractometer (Bourevestnik, St. Petersburg, Russia) with Cu Kα radiation (λ = 1.5405 Å) as an X-ray source. Scanning was carried out in the 2θ range from 10° to 50° with a scanning step of 0.1° and a data accumulation time of 5 s/step. The relative crystallinity of the polymer films was estimated using the following formula.
χ = I_C_/(I_C_ + I_A_)(1)
where I_C_ and I_A_ are the integral intensities corresponding to the crystalline and amorphous phases, respectively [35].

### 2.6. Morphology

#### 2.6.1. Optical and Polarized Light Microscopy

The surface morphology of PLA films with different OTOA contents was studied using an Olympus CX21 microscope (Olympus Corp., Tokyo, Japan) with a digital camera. The morphology of PLA crystalline spherulites was studied using polarized light microscopy on the Olympus BX51 microscope (Olympus Corp., Tokyo, Japan).

#### 2.6.2. Sorption Capacity

To determine water sorption [36], PLA film samples were cut into uniform strips of 4 × 4 cm and placed under standard atmospheric conditions (44 ± 2% RH and 20 ± 2 °C) for 24 h. After conditioning, the sample weight (m_1_) was measured. Then the samples were immersed in distilled water for 24 h at a temperature of 20 ± 2 °C to ensure uniform water sorption. Thereafter, the wet samples were hung in open air at 20 ± 2 °C for 30 min to remove excess water on the surface of the samples. Then, the weight (m_2_) of the wet samples was measured. The moisture content of the samples was calculated by the formula:%Q = (m_2_ − m_1_)/m_1_ × 100(2)
with m_1_ and m_2_ being the sample weight before and after immersion in water. The degree of hydrophilicity of the samples was determined by the water contact angle measurements in the semi-angle modification [37] for a drop of water placed on the film surface. The droplet size measurements were carried out using an OLIMPUS CX21 optical microscope (Olympus Corp., Tokyo, Japan). The results were processed using the MICAM 3.02 software (https://micam.software.informer.com, Marien van Westen).

### 2.7. Mechanical Properties

The mechanical characteristics of the samples were studied on a Zwick Z010 testing machine (ZwickRoell GmbH & Co., Ulm, Germany) at room temperature. The layout of the tested samples is shown in Figure 1.

The loading rate of the samples was 1 mm/min. During the tests, loading diagrams of the samples were recorded, namely the dependence of the load F on the deformation ε. From the diagrams obtained, tensile strength σ, elastic modulus E and elongation at break were determined. Mechanical tests were provided for five samples for each OTOA concentration, and the data were presented as mean value ± standard deviation at a significance level of *p* < 0.05.

### 2.8. Dynamic Mechanical Analysis

PLA films samples were studied by DMA under tension on a dynamic mechanical analyzer NETZSCH DMA 242 E Artemis (NETZSCH-Gerätebau GmbH, Selb, Germany). The films were cut into strips with a width of 5 mm and a length of the working part of 10 mm. The film samples were heated in the temperature range from 30 to 130 °C at a rate of 1 °C/min. During the measurement, the changes in the elastic modulus E’ and the mechanical loss tangent tgδ were recorded with increasing temperature T according to the method described in the previous works [38,39]. The glass transition temperature of the PLA films was determined from the maximum of the tgδ versus temperature dependence.

## 3. Results and Discussions

### 3.1. Film Morphology

The introduction of OTOA significantly changed the appearance and morphology of the PLA films, their structure and surface properties, as could be seen in Figure 2. The thickness of films with different OTOA concentrations did not vary significantly and was 340 ± 50 μm. It should be noted that films with high concentrations of the OTOA additive above 30% show uniform morphology and do not undergo phase separation, as is also the case with PLA/PEG blends [40]. Only the PLA film with 70% OTOA showed the signs of non-uniform morphology. Like most of the flexible-chain polymers crystallizing isothermally from the solution, PLA is characterized by a spherulitic morphology, where a spherulite is an aggregate of crystals oriented relative to a common center. A feature of spherulites is such an arrangement of primary lamellas in them that the main macromolecular chains are always perpendicular to the radius of the spherulite. The optical images of PLA films (Figure 2) clearly show the increase in the size of spherulites when OTOA is added, which could affect their physicochemical and mechanical properties. The polarized light microscopy images of PLA films are displayed in Figure 2g,h, where spherulitic morphology with a typical Maltese-cross birefringence pattern could be seen. Additionally, the increase in the size of spherulites is obvious from these images. On the other hand, optical microscopy shows that for PLA films with 50% and 70% OTOA, spherulitic morphology is almost absent. Observed increase in the size of PLA spherulites could be attributed to the plasticizing effect of OTOA, since it was shown previously that addition of plasticizer increases the spherulitic growth rate and decrease crystallization kinetics of PLA spherulites [41]. Disturbed spherulitic morphology for the PLA films with 50% and 70% OTOA could be explained by the slow crystallization kinetics and low nucleation density at high plasticizer contents, leading to hardly detectable separate spherulites.

To determine the hydrophilic-hydrophobic properties of the film surface, the water contact angle for PLA films with variable OTOA content was measured (Figure 3). In the case of pristine PLA film, good wetting of the film surface with a contact angle of 47° by water was observed (the contact angle is noticeably lower than 90°). With the introduction of OTOA in an amount of 5%, the contact angle increased to 58°, indicating additional hydrophobization of the film surface. With an increase in the amount of OTOA in the PLA matrix, a significant increase in the hydrophobicity of the surface of the PLA + OTOA films was observed. The most pronounced hydrophobic properties were observed for the PLA film with 50% OTOA content. The PLA film with 70% OTOA demonstrated a decrease in the contact angle to 68.2°, which could be explained by the presence of a large number of defects in the film.

Observed effects could be accounted for the distribution of long hydrophobic OTOA fragments in the hydrophobic regions of the PLA film and the weak interaction between them. Due to the chemical structure of PLA containing low-polarity ester groups, this biopolyester has low water solubility and belongs to moderately hydrophobic polymers [42]. At the same time, the structure of the OTOA molecule also consists mainly of hydrophobic fragments and includes, along with low-polarity ester groups, specific ozonide cycles [43,44,45]. Thus, the addition of OTOA to the PLA matrix in a wide range of concentrations (0–70%) leads to an increase in the hydrophobicity of the surface of the PLA + OTOA films.

In addition to hydrophobic properties of the film surface, water sorption capacity (Q) was studied for PLA films with variable OTOA content. As could be seen from the data presented in Figure 4, water sorption of films is about 1% and decreases with an increase in the amount of OTOA in the PLA matrix, which correlates with an increase in the water contact angle for the PLA + OTOA films (Figure 3). The PLA + OTOA film sample with the highest OTOA content (70%) demonstrates increased water sorption capacity up to 2.6%, most likely due to the non-uniform morphology and presence of defects in the film. The data obtained allows one to conclude that the addition of OTOA into the PLA matrix makes the film surface more hydrophobic, as compared to the surface of the pristine PLA film, which in turn leads to the decrease in the water sorption capacity of PLA + OTOA film materials.

### 3.2. Mechanical Properties

Changes in the morphology of modified PLA materials could have a significant effect on their mechanical properties. Therefore, the elastic modulus (E), tensile strength (σ) and relative elongation at break for the PLA + OTOA films were estimated (Figure 5a,b). Generally, the introduction of OTOA into the PLA matrix led to a decrease in both tensile strength and elastic modulus of the PLA films. The addition of 5% OTOA sharply reduced the strength characteristics of the PLA film, whereas the PLA film with 10% OTOA showed the highest values of elastic modulus and tensile strength among PLA + OTOA samples. A further increase of the OTOA content in the PLA films led to a sharp decrease in the strength characteristics of film materials. It could be assumed that the tensile strength of the films decreases due to a decrease in the degree of crystallinity for PLA films and an increase in the proportion of the plasticizing agent (OTOA) in the PLA matrix [46,47].

The effect of the OTOA addition on the relative elongation at break is somewhat different from the changes in the strength characteristics of the films described above (Figure 5c). This parameter for PLA film with 5% OTOA remains practically unchanged, as compared to the pristine PLA film, both showing poor ductility. Introduction of 10% OTOA into the PLA film led to a sharp increase in the relative elongation at the break (up to 29%), which is accompanied by a moderate decrease in the tensile strength of the material (from 29 to 21 MPa). The PLA film with 30% OTOA showed even greater value for the relative elongation at the break (52%). However, this is accompanied by a decrease by half in tensile strength. Further increase in the OTOA content above 30% was accompanied by a moderate decrease in relative elongation, while a sharp decrease in the tensile strength and elastic modulus was observed (σ = 3 MPa and E = 0.28 GPa for the PLA film containing 70% OTOA).

The observed behavior of the mechanical properties of PLA + OTOA films indicates the plasticizing effect of OTOA when it is introduced into the PLA matrix [48]. This effect could be associated with a change in the structural-dynamic state of the amorphous regions of PLA. In a recent work, it was shown that the dynamics of the amorphous phase of PLA changes significantly upon the addition of PHB [49]. The addition of PHB at concentrations of up to 30% leads to a decrease in the density of amorphous regions of PLA. A similar effect should be expected after the plasticization of PLA with OTOA molecules. The observed decrease in the relative elongation at break for the PLA films with the OTOA content above 30% could be attributed to the effect of steric hindrance, which the bulky OTOA molecule has on the segmental motion of PLA, provided that this effect exceeds the effect of plasticization. With an increase in OTOA content above 30%, the steric effect begins to prevail over the plasticization effect, increasing the PLA chain rigidity and thus reducing the relative deformation of the PLA films.

Summing up the obtained results on the mechanical properties of PLA films, it could be concluded that PLA film materials with 10% OTOA content maintain high tensile strength comparable to that of pristine PLA film and possess increased relative elongation at break. Observed combination of high elasticity and good strength characteristics obtained for PLA film materials with 10% and 30% OTOA content is important for potential applications of these materials [50].

### 3.3. Dynamic Mechanical Properties

In the DMA method, an oscillatory force is applied to a sample at a given temperature and frequency, and the material’s response to this force is measured. For viscoelastic materials such as polymers, the magnitude of the material’s response (i.e., strain amplitude) to an applied vibrational force is shifted by a phase angle δ, and the relationship between the applied stress and the strain occurring in the sample is calculated. The elastic modulus E′ indicates the ability of the sample to store or return energy. Damping or mechanical losses are usually presented as tgδ [51,52].

Figure 5a shows the changes in the elastic modulus E′ of the studied PLA films with temperature. At temperatures close to 30 °C, it could be seen that the value of the modulus increased when the OTOA content in PLA film is 10%. With an increase in the concentration of the OTOA in PLA films, a decrease in the elastic modulus is observed. The data obtained are in good agreement with the changes in the elastic modulus obtained from the mechanical tests (Figure 5a). With a further increase in temperature, a sharp decrease in the storage modulus is observed, which is associated with the transition of the material from the elastic to the highly elastic state, i.e., the glass transition of PLA film. The temperature interval of the transition for all studied PLA films is between 30 °C and 70 °C. It should be noted that PLA films with 5 and 10% OTOA are characterized by glass transition shifted to somewhat higher temperatures, as compared to glass transition for pristine PLA film. Further increase in OTOA concentration in PLA films shifts the glass transition interval to lower temperatures.

Characteristic temperatures associated with the glass transition (T_g_) in studied PLA films could be estimated from the maximum of the tgδ versus temperature dependences (Figure 6b). All samples show a single T_g_ in the range of 61–65 °C, except for the PLA film with 30% OTOA, showing significantly lower T_g_ value. The glass transition temperature of the OTOA-modified PLA films with 5 and 10% OTOA (66.8 °C and 64.7 °C) showed somewhat higher values as compared to the T_g_ for the pristine PLA film (64.2 °C). The most striking difference was observed for the PLA film with 30% OTOA, showing the T_g_ value being more than 10 °C lower than the corresponding value for the pristine PLA film. This result shows that the plasticizing effect of OTOA at concentration of 30% reaches its maximum.

As could be seen from the tgδ vs. temperature curves, the width of the temperature transition for the PLA films with 30, 50, and 70% OTOA is larger than that for the PLA films with lower OTOA concentrations. Increased width of the temperature transition evidence that PLA molecular chains exhibit a higher degree of mobility [53], which could be attributed to the plasticizing effect of OTOA.

### 3.4. FTIR Spectroscopy

To better understand the chemical interactions between PLA and OTOA, FTIR spectra of PLA + OTOA films were obtained (Figure 7a,b). The spectrum of the pristine PLA film shows characteristic bands at 1455 cm^−1^ and 1756 cm^−1^ resulting from -CH_3_ bending vibrations and C=O group stretching vibrations [54]. The absorption bands at 2944 cm^−1^ and 2995 cm^−1^ refer to the asymmetric stretching vibration of the -CH group. As could be seen, the absorption bands of PLA and OTOA are partially overlapped, which significantly complicates the analysis of the effect of OTOA addition into the PLA matrix. Though, two additional bands appear at 2927 cm^−1^ and 2856 cm^−1^ in the spectra of PLA + OTOA films, which could be attributed to the symmetric and asymmetric stretching vibrations of the -CH_2_ groups [55,56]. Since PLA does not contain -CH_2_ groups in its chemical structure, in contrast to OTOA, in which -CH_2_ group is in abundance, the presence of these bands in the FTIR spectra is the clear evidence for the inclusion of OTOA in the supramolecular structure of PLA films. As could be seen, the intensity of the absorption band at 2856 cm^−1^ shows good correlation with the OTOA content in PLA films.

In addition, the band at 1756 cm^−1^ in the pristine PLA is shifted toward lower wavenumbers (1753 cm^−1^) for PLA samples with the added OTOA. This is the evidence of weak interactions between C=O groups of PLA and polar ozonide cycles in the OTOA molecules (see below Table 1.) [57]. Based on the analysis of FTIR spectra, it could be assumed that observed interactions between polar groups in PLA and ozonide cycles in OTOA promote conformational changes associated with the reorientation of polar groups in PLA, which contributes to increased segmental mobility of PLA polymer chains [58]. Thus, low molecular weight OTOA could act as a plasticizer, affecting the mechanical and physicochemical properties of PLA film materials.

### 3.5. Differential Scanning Calorimetry

Figure 8a shows the DSC curves of the first heating for the pristine PLA film and PLA + OTOA film samples, whereas Figure 8b shows the DSC thermograms of the subsequent cooling of the samples. The characteristic endothermic peak in the first heating DSC thermograms corresponds to melting PLA (T_m_). The low-temperature step transition on the cooling thermograms in the interval of 40–60 °C could be attributed to the glass transition of PLA (T_g_). The corresponding temperatures for both transitions (T_g_ and T_m_) are shown in Table 2. The area of the melting peak was obtained, and the melting enthalpy (Δ*H*_m_) was calculated, as well as the degree of crystallinity of the PLA films according to DSC (χ). The crystallinity degree (χ) for PLA films, assuming no cold crystallization taking place during heating, was calculated according to the equation:(3)$$\chi =\frac{\Delta H_{m}}{\Delta H_{m}^{0}}\times 100\%$$
where $\Delta H_{m}$ is the experimental melting enthalpy, $\Delta H_{m}^{0}$ is the melting enthalpy value for the 100% crystalline poly(L-lactide), being 93.6 J/g [59,60]. The values of *χ* for the studied PLA film samples are given in Table 2.

The DSC thermogram of the pristine PLA film shows an endothermic peak at 169.0 °C, which is a typical melting temperature of PLA [61]. With an increase in the mass fraction of OTOA in the film, a superposition of exo- and endothermic peaks is observed in the temperature range of 140–180 °C. The endothermic peak corresponds to the melting of PLA, whereas the exothermic peak is due to a complex reaction of thermal destruction of OTOA according to the mechanism of breaking of the C-O-O-C bonds and formation of C-OH groups [62]. Previously, it was shown that this exothermic reaction is irreversible. With an increase in the OTOA content in the film, the endothermic PLA melting peak demonstrates a decrease in its area (i.e., a decrease in crystallinity) and a shift in T_m_ toward lower temperatures. The deconvolution of the overlapping exo- and endothermic calorimetric peaks (Figure 9 and Table 3) made it possible to estimate more correctly the PLA melting enthalpy and thus determine the degree of crystallinity of the studied PLA films with high OTOA contents (Table 2).

Neat PLA film was characterized by degree of crystallinity of 38.1%, as estimated by DSC, which is quite high for isothermally crystallized PLA films [63]. This could be explained by high mobility of PLA chains in the solution and high degree of supercooling, providing the thermodynamic driving force required for the growth of PLA spherulites. In addition, cold crystallization exothermic peak was not observed in DSC thermogram for the pristine PLA film and all PLA + OTOA film samples, being in contrast to the nonwoven fibrous materials based on PLA, for which the cold crystallization was previously observed [30]. This can be explained by the fact that under equilibrium crystallization conditions at ambient temperature, crystallization of neat PLA in solution proceeds with the formation of highly ordered semicrystalline structures with spherulitic morphology, which hinder further crystallization at temperatures above the glass transition due to a decrease in the segmental mobility of PLA chains. The absence of cold crystallization peaks for PLA + OTOA samples could be also attributed for the slower crystallization kinetics and decreased nucleation density due to the addition of OTOA [64].

The data shown in Table 2 show that the introduction of OTOA into PLA films led to the plasticizing effect, which was manifested in the decrease in the glass transition temperature from 57.8 to 38.1 °C and decrease in the melting temperature of PLA. The plasticizer occupies the intermolecular space between polymer chains, reducing the energy of molecular motion and the formation of hydrogen bonds between polymer chains, which in turn increases the free volume and molecular mobility [65,66,67]. As the content of the OTOA increases, the efficiency of the plasticizer in decreasing the T_g_ of PLA generally increases.

As can be seen in Table 2, both the melting temperature and the melting enthalpy decreased with the increase in the OTOA content in the PLA films. Decrease in T_m_ could be attributed to the plasticizing action of OTOA, as observed previously [30]. At the same time, a decrease in the melting enthalpy of PLA films observed upon OTOA addition evidence that along with the plasticizing effect, OTOA could impede PLA crystallization. Previously, it was shown that OTOA could hinder cold crystallization for the electrospun PLA fibers during the second heating cycle [30]. This effect could be attributed to the intermolecular interaction between the PLA terminal -OH groups and OTOA molecules observed by FTIR, leading to the decrease in the mobility of PLA polymer chains. This provides the physical hindrance for PLA crystallization and leads to the decrease in the crystallinity of PLA films after increase in OTOA content.

### 3.6. X-ray Diffraction Analysis

To independently confirm the effect of OTOA on the crystal characteristics of PLA + OTOA films, the samples were studied by X-ray diffraction (XRD) and the values of the degree of crystallinity of the films were obtained. The X-ray diffraction patterns of PLA films with different OTOA contents are shown in Figure 10. Pristine PLA film showed the main diffraction peaks at 2θ angles of 16.2° and 18.6°, confirming the presence of PLA crystal structures in the film. PLA has a strong diffraction at 16.2°, related to the crystalline α phase of PLA [68]. As could be seen, PLA and PLA + OTOA films have the same crystal structure. An increase in the mass fraction of OTOA does not lead to a significant change in the position of the diffraction peaks corresponding to the PLA crystalline phase. On the other hand, it leads to an increase in the contribution of the amorphous phase (amorphous halo), corresponding to the OTOA phase and/or amorphous regions of the semicrystalline PLA structure.

The degree of crystallinity of PLA films was estimated by the formula given above (see materials and methods) and the corresponding values are given in Table 2. A noticeable decrease in the crystallinity of the system is observed with an increase in the mass fraction of OTOA in PLA films. For the film with the OTOA content of 70%, the crystallinity of the system drops up to 14.3%, being nonetheless quite high [41]. The crystallinity values estimated by the XRD method correlate well with the corresponding values obtained from the DSC data. The only exception is the PLA film with 70% OTOA content, for which estimation of the χ value from DSC data is complicated by the large overlapping calorimetric peaks.

Summing up the obtained results, it could be concluded that the morphology, sorption capacity, as well as thermodynamic and mechanical properties of PLA + OTOA films could be controlled by the OTOA content in the films. Thus, PLA film sample containing 10% OTOA is characterized by a combination of high elasticity and good strength characteristics (tensile strength 21 MPa, relative elongation at break 26%). The sample with the addition of 30% OTOA gives an even greater increase in relative elongation (up to 52%) with a significant loss of tensile strength. The combination of high elasticity and strength is of great importance for the various applications of PLA film materials. FTIR spectroscopy revealed an interaction between the OTOA molecules, which act as a plasticizer, and the PLA matrix. The plasticizing effect of OTOA, as shown by DSC, DMA, and mechanical tests, contributed to the changes in the thermal and mechanical properties of PLA films modified with OTOA. At the same time, at high OTOA concentrations (>30%), the effects related to the interaction of OTOA with PLA polymer chains and the hindrance of the segmental motion of PLA polymer ends could prevail over the plasticizing effect of OTOA, leading to the decrease in crystallinity of PLA films, deterioration of the morphology and mechanical properties of PLA films. It could be concluded that the PLA films with 10% and 30% OTOA possess optimal physicochemical properties, such as improved morphology, mechanical properties, and water sorption characteristics. Since OTOA exhibits a pronounced antimicrobial activity [31,62,69], it could be used not only as an efficient plasticizer, but also as a functional antimicrobial additive. Thus, developed PLA + OTOA films have a high potential to be used as biodegradable materials with antibacterial activity, for example, as antimicrobial packaging materials, which could provide protection from physical, chemical, and microbiological factors and reduce the negative impact on the environment.

## 4. Conclusions

In this article, the morphological, physicochemical, mechanical, and thermal properties of PLA films after the addition of different concentrations of oleic acid triglyceride ozonide (OTOA) were studied. Analysis of the film surface showed changes in the film morphology as a result of the addition of OTOA, which were manifested in an increase in the size of spherulites. PLA films containing 10% and 30% OTOA exhibited optimal mechanical and elastic properties, combining high elasticity and good tensile strength. The obtained results are important for potential packaging applications of the studied PLA film materials. Contact angle measurements showed that OTOA addition leads to significant hydrophobization of PLA films, whereas FTIR spectroscopy revealed weak interactions between OTOA and the PLA matrix. It was found that OTOA acts as a plasticizer and leads to an increase in PLA segmental mobility, which in turn contributes to changes in the thermodynamic and mechanical properties of PLA films. The results of DSC and XRD showed that OTOA promotes the process of PLA amorphization, therefore reducing the crystallinity of the resulting PLA + OTOA film materials. Eventually, the obtained results evidence that the morphological, thermodynamic, and mechanical properties of PLA + OTOA films could be controlled by the OTOA content in the films. Since OTOA possess pronounced antimicrobial activity, studied composite PLA + OTOA films could provide additional functionality as compared to biodegradable packaging materials based on pristine PLA. Developed PLA film materials with the optimal OTOA content of 10% and 30% could be used in various packaging and biomedical applications.

## Data Availability

Not applicable.

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
