# Peer review of "Improvement of the Structure and Physicochemical Properties of Polylactic Acid Films by Addition of Glycero-(9,10-trioxolane)-Trialeate"

_polymers, 2022, doi:10.3390/polym14173478_

Round 1

Reviewer 1 Report

In this manuscript, the authors studied the morphological, physicochemical, mechanical, and thermal properties of PLA films after the addition of different concentrations of oleic acid triglyceride ozonide (OTOА).  The morphological analysis of the films showed that the addition of OTOA increased the diameter of PLA spherulites and, as a consequence, increased the proportion of amorphous regions in PLA films. A study of the thermodynamic properties of PLA films by differential scanning calorimetry (DSC) demonstrated a decrease in the glass transition temperature of the films with an increase in the OTOA content. The results of DSC and XRD 495 showed that OTOA promotes the process of PLA amorphization, thereby reducing the crystallinity of the resulting PLA + OTOA film materials. According to DSC and XRD data, the degree of crystallinity of the PLA films showed a tendency to decrease with an increase in the OTOA content in the films, which could be accounted for the plasticizing effect of OTOA. Developed PLA film materials with the optimal OTOA content of 10% and 30% could be used in various packaging and biomedical applications. This study is good and important to nanocomposite materials preparation and characterization which determine unambiguously how addition can influence the fracture behaviors of the PLA film. The interpretations and conclusions are well justified by the results. In addition, the quantity and quality of the figures are appropriate. We believe that this research subject is promising for developing film composite high mechanical compared to a virgin film.

Summary: I recommend publishing this manuscript after considering my comments on the attached file.

Reviewer 2 Report

Glycero-(9,10-trioxolane)-trioleate (ozonide of oleic acid triglyceride, OTOA) was introduced into polylactic acid (PLA) films in amounts of 5, 10, 30, 50, and 70% w/w. The morphological, mechanical, thermal and water absorption properties of PLA films after the OTOA addition have  been studied. The manuscript need minor revise before publish.

(1) Please provide the molecular structure of the OTOA as a figure.

(2) Figure 7. The infrared spectrum with the infrared absorption peak downward is not common used.

(3) In general, the first heating in the DSC program belongs to the elimination of thermal history and cannot be used as data. Because of this, the DSC peak shape in Figure 8A makes me feel very strange
